# Haplotype Analysis of GmSGF14 Gene Family Reveals Its Roles in Photoperiodic Flowering and Regional Adaptation of Soybean

**DOI:** 10.3390/ijms24119436

**Published:** 2023-05-29

**Authors:** Liwei Jiang, Peiguo Wang, Hongchang Jia, Tingting Wu, Shan Yuan, Bingjun Jiang, Shi Sun, Yuxian Zhang, Liwei Wang, Tianfu Han

**Affiliations:** 1College of Agriculture, Heilongjiang Bayi Agricultural University, Daqing 163316, China; jiangliwei827@163.com (L.J.); 13836962211@126.com (Y.Z.); 2MARA Key Laboratory of Soybean Biology (Beijing), Institute of Crop Sciences, The Chinese Academy of Agricultural Sciences, 12 Zhongguancun South Street, Beijing 100081, China; wpg15101224039@163.com (P.W.); jiahongchangheihe@163.com (H.J.); wutingting@caas.cn (T.W.); yuanshan@caas.cn (S.Y.); jiangbingjun@caas.cn (B.J.); sunshi@caas.cn (S.S.); 3Department of Crop Genetics and Breeding, College of Agronomy, Gansu Agricultural University, Lanzhou 730070, China; 4Heihe Branch, Heilongjiang Academy of Agricultural Sciences, Heihe 164399, China

**Keywords:** soybean, *GmSGF14* gene family, photoperiodic flowering, haplotype analysis, geographical adaptation

## Abstract

Flowering time and photoperiod sensitivity are fundamental traits that determine soybean adaptation to a given region or a wide range of geographic environments. The General Regulatory Factors (GRFs), also known as 14-3-3 family, are involved in protein–protein interactions in a phosphorylation-dependent manner, thus regulating ubiquitous biological processes, such as photoperiodic flowering, plant immunity and stress response. In this study, 20 soybean *GmSGF14* genes were identified and divided into two categories according to phylogenetic relationships and structural characteristics. Real-time quantitative PCR analysis revealed that *GmSGF14g*, *GmSGF14i*, *GmSGF14j*, *GmSGF14k*, *GmSGF14m* and *GmSGF14s* were highly expressed in all tissues compared to other *GmSGF14* genes. In addition, we found that the transcript levels of *GmSGF14* family genes in leaves varied significantly under different photoperiodic conditions, indicating that their expression responds to photoperiod. To explore the role of *GmSGF14* in the regulation of soybean flowering, the geographical distribution of major haplotypes and their association with flowering time in six environments among 207 soybean germplasms were studied. Haplotype analysis confirmed that the *GmSGF14m^H4^* harboring a frameshift mutation in the 14-3-3 domain was associated with later flowering. Geographical distribution analysis demonstrated that the haplotypes related to early flowering were frequently found in high-latitude regions, while the haplotypes associated with late flowering were mostly distributed in low-latitude regions of China. Taken together, our results reveal that the *GmSGF14* family genes play essential roles in photoperiodic flowering and geographical adaptation of soybean, providing theoretical support for further exploring the function of specific genes in this family and varietal improvement for wide adaptability.

## 1. Introduction

Soybean (*Glycine max*) originated from the temperate region of China and is a typical short-day (SD) plant that accelerates the transition from vegetative to reproductive growth under SD conditions [1,2,3]. Photoperiodic flowering is a critical factor in determining the regional adaptability and productivity of soybean [4,5]. The major genes, including *E1* [6], *E2* [7], *E3* [8], *E4* [9], *GmFTs* [10,11,12,13,14,15,16,17,18], *J* [19,20,21], *LUX* [20], *GmLHYs* [22,23], *GmPRR3a* [24] and *GmPRR37/GmPRR3b* [24,25], were well characterized as participating in the photoperiod control of flowering. As an ancient paleopolyploid species, soybean experienced genome duplications, resulting in nearly 75% of the genes being present in multiple copies [26]. For example, 10 *GmFTs* were found in soybean [10], including the flowering-promoting FT homologues *GmFT2a* [10,11,12,13], *GmFT5a* [10,11,12,13,14], *GmFT2b* [17] and the flowering-inhibiting FT homologues *GmFT1a* [11,15] and *GmFT4* [16]. Molecular characterization of photoperiodic flowering in soybean is complicated due to the functional redundancy and divergence of gene families. Therefore, investigating the gene family comprehensively and systematically will provide a foundation for further research on the regulatory mechanisms of soybean flowering.

14-3-3 proteins were originally discovered in cow brain by Moore and Perez [27], and are highly conserved eukaryotic proteins [28]. In *Arabidopsis*, there are 13 members in the 14-3-3 gene family [29], which are designated as either G-box Factors (GFs) or General Regulatory Factors (GRFs) [30]. The 14-3-3 proteins can form homodimer or heterodimers on N-terminal regions, which allow the dimer to combine two target proteins simultaneously. They function as regulators in signal transduction through binding to phosphoserine-containing proteins [31,32,33,34,35]. The 14-3-3 proteins are involved in various physiological processes, including primary metabolism [36,37,38], hormone regulation [39,40,41], and response to biotic/abiotic stress [42,43]. 

Increasing evidence indicates that *GF14* genes also play crucial roles in flowering regulation of plants. For instance, in tomato, 14-3-3/74 interacts with SP (SELF-PRUNING), which is an ortholog of *TFL1* (*TERMINAL FLOWER1*), and maintains the indeterminate state of inflorescence in *Arabidopsis* [44,45,46]. Overexpression of 14-3-3 genes in tomato was previously shown to partially complement the phenotype of *sp* mutation [45]. In rice, 14-3-3 proteins mediate the interaction between FT (FLOWERING LOCUS T) and FD to form the florigen-activated complex (FAC), which regulates flowering and induces inflorescence development [47,48,49]. GF14c also acts as a negative regulator of flowering through interacting with Hd3a [50]. Similarly, most GF14 proteins in *Arabidopsis* can interact with FT and TFL [51]. In cotton, GhGRF3/6/9/14/15 can interact with GhFT and GhFD to form a FAC; transgenic plants overexpressing *GhGRF3/6/9/15* display late flowering in *Arabidopsis*, while *GhGRF14*-overexpressing plants show early flowering [52]. In soybean, 14-3-3 proteins are involved in nodulation [53] and isoflavone synthesis [54]; however, there is very limited knowledge of their roles in flowering regulation.

The objective of this study was to investigate the molecular and evolutionary characteristics of the soybean *GmSGF14* gene family, as well as its role in flowering regulation under different photoperiods. We identified 20 soybean *GmSGF14* genes and analyzed their expression patterns under different photoperiodic conditions. Natural variations in *GmSGF14* alleles were detected across 207 re-sequenced soybean varieties. Furthermore, we examined the geographical distribution of major haplotypes and their association with flowering time in six different environments. Our findings provided insights into *GmSGF14* gene family roles in flowering regulation of soybean.

## 2. Results

### 2.1. Identification and Analysis of the Physicochemical Properties of the GmSGF14 Gene Family in Soybean

To identify the *GmSGF14* genes in soybean, we performed a BLAST search using the 14-3-3 protein of *Arabidopsis* as queries via TBtools. Finally, 20 genes were identified and annotated as being *GmSGF14* genes based on the complete 14-3-3 domain, before being named as *GmSGF14a*–*GmSGF14t* (Appendix A). The chromosomal localization analysis of *GmSGF14* genes showed that they were unevenly distributed in 13 of the 20 soybean chromosomes (Figure 1A). Further collinearity analysis revealed numerous fragment repetitions among the *GmSGF14* gene family, indicating functional similarity among *GmSGF14* genes (Figure 1A). 

Gene characteristics, including the length of the protein sequence, the protein molecular weight (MW), the isoelectric point (pI), the instability index and the aliphatic index, were analyzed (Appendix A). The results showed the *GmSGF14* genes encoded proteins with amino acid numbers ranging from 71 aa (GmSGF14t) to 315 aa (GmSGF14k), while MW ranged from 7.92 kDa to 35.2 kDa. The pI of the proteins ranged from 4.67 (GmSGF14g, GmSGF14h) to 5.70 (GmSGF14t), and the instability coefficient varied from 32.3 (GmSGF14t) to 53.8 (GmSGF14r). All of the GmSGF14 proteins were hydrophilic proteins (GRAVY < 0). 

### 2.2. Phylogenetic Analysis of the GmSGF14 Gene Family in Soybean

To further explore the evolutionary relationship and classification of the *GmSGF14* gene family, a phylogenetic tree was constructed based on the multiple sequence alignment of 14-3-3 protein sequences of *Arabidopsis*, rice and soybean. The phylogenetic analysis indicated that all the 14-3-3 members were classified into two subfamilies: ε class and non-ε class (Figure 1B). Among the 20 GmSGF14 proteins, 11 belong to the ε class group (GmSGF14c, d, e, f, l, n, o, p, q, r and t), and 9 belong to non-ε class group (GmSGF14a, b, g, h, i, j, k, m, and s). Furthermore, we found that genes on the same chromosome, such as *GmSGF14g* and *GmSGF14f*, and *GmSGF14l* and *GmSGF14m,* were not divided into the same group (Figure 1). These results indicate that *GmSGF14* genes experienced evolutionary divergence and functional diversity.

### 2.3. Gene Structure, Motif Composition and Promoter Characterization of the GmSGF14 Gene Family

The exon–intron structure of all the identified *GmSGF14* genes was examined to gain more insight into the evolution of the 14-3-3 family in soybean. *GmSGF14* genes in the non-ε group had a maximum of four introns, while the number of exons ranged from two (*GmSGF14s*) to five (*GmSGF14k*). In the ε group, *GmSGF14* genes contained four to six introns and six exons, except for *GmSGF14t*, which had only one intron and two exons (Figure 2B). The results indicated that different *GmSGF14* genes diverged structurally during evolution. Subsequently, 10 conserved motifs of *GmSGF14* were identified through Multiple Em for Motif Elicitation (MEME). As shown in Figure 2C, Motif 6 was distributed in all *GmSGF14* genes, and seven Motifs (Motif 1-7) constituting the 14-3-3 domain were highly conserved. *GmSGF14* members within the same groups were usually found to share a similar motif composition. In addition, motifs 8 and 10 are unique to the non-ε group, and motif 9 is specific to the ε group.

In order to explore the potential expression regulation patterns of the *GmSGF14* genes, cis-elements were predicted in the 2 kb sequence upstream of these genes (Appendix A). A total of 37 cis-acting elements involved in plant growth and development were identified, including elements involved in light response, endosperm expression, cell cycle regulation, meristem expression, circadian rhythm regulation, phytohormone response and stress response (Appendix A). The number of light-response cis-acting elements made up a significant percentage in the promoter regions of 20 *GmSGF14* genes. Therefore, we speculated that *GmSGF14* may play an important role in growth and development, especially in photoperiodic flowering.

### 2.4. Expression Patterns of GmSGF14 and Its Response to Different Photoperiods

We examined the expression levels of *GmSGF14* genes in different tissues (root, hypocotyl, stem, unifoliolate leaf, trifoliolate leaf, and shoot apex) of Zhonghuang 13 (ZH13), which is a soybean variety widely grown in China, under different photoperiod treatments (Figure 3). Seven genes (*GmSGF14h*, *GmSGF14n*, *GmSGF14o*, *GmSGF14p*, *GmSGF14q*, *GmSGF14r* and *GmSGF14t*) were hardly expressed in all treatments. In contrast, *GmSGF14g*, *GmSGF14i*, *GmSGF14j*, *GmSGF14k*, *GmSGF14m* and *GmSGF14s* were highly expressed in all tissues. The leaf is the major organ that responds to photoperiod to induce flowering. Thus, we compared the expression of *GmSGF14* genes in leaf under LD and SD conditions. We found that under SD conditions, compared with LD conditions, the expression levels of *GmSGF14h*, *GmSGF14m*, *GmSGF14n*, *GmSGF14o*, *GmSGF14p* and *GmSGF14q* were higher in unifoliolate leaves and trifoliolate leaves, while *GmSGF14a*, *GmSGF14b* and *GmSGF14j* showed higher expression only in trifoliolate leaves. Under LD conditions, *GmSGF14g and GmSGF14i* were highly expressed in the unifoliolate leaves, and the expression of *GmSGF14s* was higher in unifoliolate and trifoliolate leaves compared to SD conditions. *GmSGF14c*, *GmSGF14d*, *GmSGF14e*, *GmSGF14f*, *GmSGF14k*, *GmSGF14l*, *GmSGF14r* and *GmSGF14t* were more highly expressed in the unifoliolate leaves and less expressed in the trifoliolate leaves under LD conditions compared to SD conditions. These results showed that *GmSGF14* genes have different expression patterns and responses to photoperiod, which indicated that *GmSGF14* genes might be involved in soybean photoperiodic flowering.

### 2.5. Haplotype Analysis of 20 Soybean GmSGF14 Family Genes in Soybean Germplasm with Diverse Geographical Origins

To evaluate the effect of mutations in *GmSGF14* genes on soybean adaptation, we examined the genotypes of 207 re-sequenced soybean varieties. Further geographical distributions regarding major haplotypes and their association with flowering time in six environments were conducted (Figure 4 and Figure 5). In general, haplotypes of the *GmSGF14* genes exhibited more diversity in the varieties from China than those from the US, suggesting that the country of origin for cultivated soybean is China (Figure 5).

For *GmSGF14a*, 15 SNPs and 2 Indels were detected (Appendix A), and 6 haplotypes were identified. *GmSGF14a^H2^*, *GmSGF14a^H3^* and *GmSGF14a^H4^* showed no significant difference in flowering time and were distributed in all regions, while *GmSGF14a^H5^* exhibited significantly later flowering and was only distributed in the south of China (SC) (Figure 4A and Figure 5A). The results indicated that *GmSGF14a^H5^* could better adapt to short-day environments and might facilitate soybean genetic improvement in low-latitude regions such as Southern China (SC).

Four SNPs were located in the 5’UTR and intron regions, and three haplotypes were discovered for *GmSGF14b* (Appendix A). *GmSGF14b^H3^* was distributed in Huang–Huai–Hai Rivers Valley Region (HHH) and Northeastern China (NE), while *GmSGF14b^H2^* covered a large proportion of the haplotypes found across China and the US. *GmSGF14b^H3^* exhibited significantly later flowering in Heihe; however, it showed no significance in the five other environments (Figure 4B and Figure 5B).

Based on 14 SNPs and 4 Indels, *GmSGF14c* was divided into 6 haplotypes (Appendix A). *GmSGF14c^H1^* showed earlier flowering, and the frequency of *GmSGF14c^H1^* decreased with decreasing latitude in China (Figure 4C and Figure 5C). With the increase in latitude in China, the proportion of *GmSGF14c^H3^* increased (Figure 5C).

For *GmSGF14e*, we identified 10 SNPs and 1 Indel, and defined 3 haplotypes (Appendix A). *GmSGF14e^H1^* flowered significantly earlier than *GmSGF14e^H2^* and *GmSGF14e^H3^* in all six environments, except for Heihe (Figure 4D). With increasing latitude in China, the proportion of *GmSGF14e^H1^* increased, whereas the proportion of *GmSGF14e^H2^* and *GmSGF14e^H3^* decreased (Figure 5D).

Five haplotypes were identified based on seven SNPs and five Indels of *GmSGF14f* (Appendix A). It was found that *GmSGF14f^H1^* was mostly distributed in NE and the US, with significantly earlier flowering compared with *GmSGF14f^H2^*, *GmSGF14f^H3^* and *GmSGF14f^H4^* (Figure 4E and Figure 5E). *GmSGF14f^H5^* showed significantly later flowering among plants distributed in HHH and SC than those found in the NE and the US (Figure 4E and Figure 5E). Further analysis of these two unique haplotypes (*GmSGF14f^H1^* and *GmSGF14f^H5^*) may contribute to the genetic improvement of soybean in SC and NE.

Three SNPs and three haplotypes were identified for *GmSGF14h* (Appendix A). *GmSGF14h^H2^* was the haplotype most widely distributed across China and the US (Figure 5F), and there was no significant difference in flowering times among the three haplotypes (Figure 4F). 

We found one SNP and two Indels in *GmSGF14i*, and identified three haplotypes (Appendix A). Among the three haplotypes, *GmSGF14i^H1^* associated with significantly earlier flowering was mainly distributed in NE and the US. *GmSGF14i^H2^* was the most abundant (Figure 5G), while *GmSGF14i^H3^* flowered later and tended to distribute in HHH and SC (Figure 4G and Figure 5G).

We found 47 SNPs and two haplotypes in *GmSGF14k* (Appendix A). In six environments, *GmSGF14k^H2^* only showed significantly earlier flowering in Xinxiang compared to *GmSGF14k^H1^* (Figure 4H) and was the most widely distributed haplotype across China and the US (Figure 5H).

For *GmSGF14m*, a total of 4 haplotypes were identified based on 10 SNPs and 6 Indels (Appendix A). *GmSGF14m^H4^* carried a frameshift mutation that resulted in partial deletion of 14-3-3 domain in the encoded protein (Appendix A). Compared to *GmSGF14m^H1^*, *GmSGF14m^H4^* showed significantly later flowering and the percentage of *GmSGF14m^H1^* increased with the increase in latitude in China, while *GmSGF14m^H4^* was had the opposite characteristics (Figure 4I and Figure 5I). These results suggest that the frameshift mutation carried out in *GmSGF14m^H4^* leading to the loss of 14-3-3 domain may result in late flowering.

Based on six SNPs, two haplotypes were identified for *GmSGF14n* (Appendix A). *GmSGF14n^H1^*, which was primarily distributed at high latitudes, flowered significantly earlier than *GmSGF14n^H2^* (Figure 4J and Figure 5J). The percentage of *GmSGF14n^H2^* increased with the decrease in latitude in China (Figure 5J).

One SNP and one Indel were found in *GmSGF14o*, and three haplotypes were identified (Appendix A). Through comparing the flowering time of the three haplotypes in the six environments in China, we found that *GmSGF14o^H3^* flowered significantly later (except Sanya and Xiangtan), and the phenotypic difference tended to become more pronounced with increasing latitude (Figure 4K). The frequencies of *GmSGF14o^H3^* varied across regions, and no variety found in the US harbored this haplotype (Figure 5K).

For *GmSGF14p*, three haplotypes were defined according to seven SNPs, and SNP-Chr13:37266730 and SNP-Chr13:37269199 were missense mutations ((Asp/Glu) and (Lys/Ile)) (Appendix A). Compared to *GmSGF14p^H1^* and *GmSGF14p^H2^*, *GmSGF14p^H3^* showed significantly later flowering and was mostly distributed in HHH and SC (Figure 4L and Figure 5L).

Based on 12 SNPs and 4 Indels, *GmSGF14q* was divided into 3 haplotypes (Appendix A). The percentage of *GmSGF14q^H1^* was high in NE, and *GmSGF14q^H2^* and *GmSGF14q^H3^* showed later flowering and were mainly found in HHH and SC (Figure 4M and Figure 5M).

For *GmSGF14r*, 25 SNPs and 3 Indels were discovered, and SNP-Chr20:2850044 was located at the 14-3-3 domain, resulting in an amino acid substitution (Glu/Asp) (Appendix A). Eight haplotypes were identified and were related to different genetic effects on flowering under different environments (Figure 4N). *GmSGF14r^H4^* was the haplotype most widely distributed across China and the US (Figure 5N). The rich genetic diversity of *GmSGF14r* may contribute to the varied phenotypic effects on soybean flowering across different photoperiods.

Among the five SNPs in *GmSGF14s*, SNP-Chr17:34108428, SNP-Chr17:34108738 and SNP-Chr17:34108786 were missense mutations that resulted in amino acid substitutions (Appendix A). We identified two haplotypes, of which *GmSGF14s^H2^* was mainly distributed in HHH and only showed significantly later flowering in Heihe when compared to the other five environments. (Figure 4O and Figure 5O).

## 3. Discussion

In plants, previous studies showed that 14-3-3 proteins are involved in various biological processes, including photoperiodic flowering [44,45,55], plant immunity [55,56] and stress responses [57,58]. However, comprehensive analysis of the 14-3-3 family and their functions in soybean flowering is limited. In this study, a total of 20 *GmSGF14* genes were identified and classified into ε and non-ε classes via phylogenetic analysis (Figure 1B), which was consistent with model plants, such as *Arabidopsis* [59,60,61] and rice [62]. This discovery demonstrated that 14-3-3 genes experienced expansion in soybean, compared to 15 genes in *Arabidopsis* and 8 genes in rice. Chromosomal distribution and synteny analysis confirmed that gene and segmental duplication events played important roles in the expansion of *GmSGF14* gene family (Figure 1A), supporting the complex history of whole genome duplications in soybean [26,63,64]. Conserved protein motifs analysis further demonstrated that ε and non-ε groups possess different motif structures (Figure 2), indicating diverse functions in accordance with 14-3-3 proteins in Arabidopsis and rice. Expression profiles revealed that soybean *GmSGF14* genes in all examined organs showed differential expression, and their expression in leaf and shoot apex varied significantly under different light conditions, suggesting their involvement in photoperiodic regulation of soybean flowering (Figure 3). 

Previous studies confirmed the crucial role of 14-3-3 family genes in flowering regulation [44,45,46,47,48,49,50,51,52]. The activation of floral initiation in the shoot apical meristem (SAM) is triggered through the florigen activation complex (FAC). In rice, 14-3-3 proteins can form a FAC with florigen protein Hd3a and bZIP transcription factor OsFD1. The complex further upregulates the transcription of rice *APETALA1*(*AP1*) homologue *OsMADS15*, inducing flowering [47]. Moreover, the FAC can also be synthesized in rice leaves, where *RCN* (*RICE CENTRORADIALIS*), which is a homologue of *Arabidopsis TFL1*, forms a florigen repression complex (FRC) through competition with Hd3a for 14-3-3 and OsFD1 binding [65]. The balance between FAC and FRC fine-tunes the florigen activity to ensure flowering at appropriate times [48]. In *Arabidopsis* [51], potato [45] and cotton [52], 14-3-3 was reported to interact with FT or FD to form complexes. In soybean, 10 *GmFT* homologs were identified [10]. *GmFT2a*/*2b*/*3a*/*5a* promote flowering [10,11,12,13,14,17], while *GmFT1a*/*4* repress flowering [15,16]. Both GmFT2a and GmFT5a interact with GmFDL19, upregulating downstream flowering-related genes, which indicates that the FAC is conserved in soybean [12]. Currently, the understanding of 14-3-3 in soybean flowering is limited; thus, the 20 GmSGF14 proteins identified in our study will facilitate the characterization of the FACs in soybean flowering regulation.

The cultivated soybean originated in the temperate region of China and was planted worldwide as an important economic crop due to its high protein and oil content [66]. This wide distribution can be attributed to the rich natural variations in and combinations of genes controlling flowering times [67,68]. Natural variation in *GmELF3* confers long juvenility and improves soybean adaptation in the tropics [19]. In contrast, natural variations in *GmPRR37* (*GmPRR3b*) affect photoperiodic flowering and contribute to soybean adaptation in high-latitude regions [24,25]. To evaluate the effects of variations in *GmSGF14* on soybean flowering and adaptation, we investigated the genotypes of 207 varieties collected across China and the US, and analyzed the flowering-time phenotypes and geographical distributions of the major haplotypes. We found that *GmSGF14m^H4^* carries a single-base deletion that results in a frameshift mutation and premature termination of the encoded protein (Appendix A). This null mutant was significantly associated with late flowering, and its frequency increased with decreasing latitude in China (Figure 4I and Figure 5I). These results indicate that *GmSGF14m* might function as a flowering promoter, while the frameshift mutation may lead to late flowering. Additionally, we found that *GmSGF14c^H1^*, *GmSGF14e^H1^*, *GmSGF14f^H1^*, *GmSGF14i^H1^*, *GmSGF14m^H1^*, *GmSGF14n^H1^*, *GmSGF14p^H1^*, *GmSGF14p^H2^*, *GmSGF14q^H1^* and *GmSGF14r^H2^* were associated with early flowering and primarily distributed in higher latitudes of China. On the other hand, *GmSGF14a^H5^*, *GmSGF14e^H2^*, *GmSGF14e^H3^*, *GmSGF14f^H5^*, *GmSGF14i^H3^*, *GmSGF14m^H4^*, *GmSGF14n^H2^*, *GmSGF14o^H3^*, *GmSGF14p^H3^*, *GmSGF14q^H3^* and *GmSGF14r*^H5^ were related to late flowering and primarily distributed in lower latitude regions. We speculate that the diverse genetic variation in GmSGF14 contributed to soybean cultivation across different latitudes. Developing Kompetitive Allele Specific PCR (KASP) markers and gene chips for these variations can provide information about the molecular breeding of soybean flowering times. Further functional characterization of *GmSGF14* family genes in soybean could provide opportunities to utilize genome-editing tools to modify the functional status of *GmSGF14* family members, thus facilitating precise prediction of flowering times in soybean genetic improvement.

## 4. Materials and Methods

### 4.1. Plant Materials, Treatments and Multiple-Site Experiments

For the expression pattern analysis of *GmSGF14* family genes, a widely grown soybean variety Zhonghuang 13 (ZH13) was grown in a controlled culture room at 26 °C under short-day (SD, 12 h: 12 h, light: dark) and long-day (LD, 16 h: 8 h, light: dark) conditions. After entraining for 14 days, root, hypocotyl, stem, unifoliolate leaf, trifoliolate leaf and shoot apex were sampled. Samples were collected after 4 h exposure to light.

A total of 207 re-sequenced soybean germplasms were used for haplotype analysis [69], including 97 from Northeast China (NE), 46 from Huang–Huai–Hai Rivers Valley Region (HHH), 37 from South China (SC), and 27 from the USA (Appendix A). The panel was planted in six regions: Sanya (18°18′ N, 112°39′ E), Xinxiang (35°08′ N, 113°45′ E) and Beijing (40°13′ N, 116°33′ E) in 2016, and Xiangtan (27°40′ N, 112°39′ E), Changchun (43°50′ N, 124°82′ E) and Heihe (50°15′ N, 127°27′ E) in 2017; the regions were named SY2016, BJ2016, XX2016, XT2017, CC2017 and HH2017, respectively [17]. Flowering times for 207 soybean varieties in six environments were recorded as days from the emergence to the R1 stage (the time at which the first flower opens at any node on the main stem) [70] and determined through taking the average of two replicates.

### 4.2. Identification, Bioinformatic and Phylogenetic Analysis of GmSGF14

The 14-3-3 protein sequences of *Arabidopsis thaliana* and rice were downloaded from TAIR (https://www.arabidopsis.org/) (accessed on 12 September 2022) and the Phytozome database (https://phytozome-next.jgi.doe.gov/) (accessed on 12 September 2022) and were used as queries for searching for soybean homologous genes. Soybean genome annotation was downloaded from the Phytozome 13.0 database (https://phytozome-next.jgi.doe.gov/) (accessed on 12 September 2022). TBtools was used to search GmSGF14 members from soybean genome database with an E-value threshold of <1 × e^−10^ [71]. All candidate genes were further confirmed to contain the 14-3-3 domain (PF00244) using Pfam (http://pfam.xfam.org/) (accessed on 14 September 2022) and the SMART program (http://smart.embl-heidelberg.de/) (accessed on 14 September 2022).

Duplications of *GmSGF14* family genes were analyzed via the Multiple Collinearity Scan toolkit (MCScanX) with the default parameters, and a visual synchronous analysis diagram was constructed using TBtools [71]. The amino acid numbers, molecular weight (MW), isoelectric point (pI), instability coefficient, fat coefficient and hydrophilicity of identified GmSGF14 proteins were analyzed via ExPASy ProtParam (https://web.expasy.org/protparam/) (accessed on 28 September 2022). 

MEGA 7.0 software was used to perform multiple sequence alignment on the reported 14-3-3 protein sequences of *Arabidopsis thaliana*, rice and soybean, as well as to construct the phylogenetic tree with the Neighbor-Joining (NJ) method. The bootstrap value was set to 1000. Bootstrap resampling (100) was used to assess the reliability of interior branches, and other parameters were default values. 

### 4.3. Sequence Analysis of Soybean GmSGF14 Family Genes

The exon–intron pattern of *GmSGF14* gene family was analyzed using TBtools v1.0+ software through inputting gene annotation GFF files. The online program MEME (https://meme-suite.org/meme/tools/meme) (accessed on 9 October 2022) was used to identify the conserved motif of GmSGF14 proteins, and the maximum number of motifs was set to 10. To identify the cis-elements, TBtools was used to extract a 2 kb genomic sequence upstream from the start codon (ATG) of the *GmSGF14* family genes gathered from the soybean genome database, and PlantCARE (http://bioinformatics.psb.ugent.be/webtools/plantcare/html/) (accessed on 21 October 2022) was used to predict the cis-acting elements of the promoter. Finally, TBtools v1.0+ software was used for visual mapping [71].

### 4.4. Expression Profile Analysis of Soybean GmSGF14 Gene Family

The total RNA of different tissues of ZH13 was extracted using Easy Fast Plant Tissue Kit (TianGen, Beijing, China), and RNA was reverse transcribed into cDNA with the FastKing RT Kit (With gDNase) (TianGen, Beijing, China). Primers for qRT-PCR were designed using the NCBI (https://www.ncbi.nlm.nih.gov/tools/primer-blast/) (accessed on 14 December 2022) (Appendix A). Using ABI QuantStudio^TM^ 7 flex (Applied Biosystems, San Francisco, CA, USA), qRT-PCR was performed with Taq Pro Universal SYBR qPCR Master Mix (Vazyme, Nanjing, China), and each sample contained three biological replicates. *GmActin* (*Glyma18g52780*) was used as the internal reference, and the relative expression was calculated via the 2^−∆∆Ct^ method.

### 4.5. Haplotype and Correlation Analysis of Soybean GmSGF14 Gene Family

The natural variation in *GmSGF14* genes was retrieved from the NCBI database under Short Read Archive (SRA) Accession Number SRP062560 and PRJNA589345 [17,69], and the haplotype and data processing analyses were performed using TASSEL 5 and EXCEL. We used GraphPad Prism 8 to analyze the association between *GmSGF14* haplotypes and flowering time via Duncan’s multiple range test with *p* < 0.05 as the significant level.

## Figures and Tables

**Figure 1 ijms-24-09436-f001:**
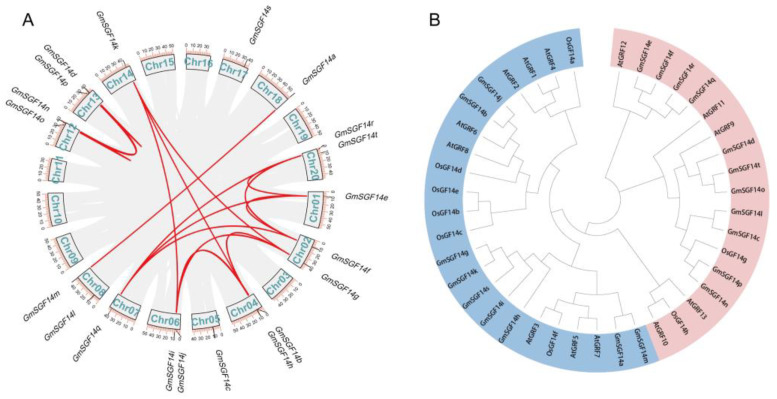
Syntenic and phylogenetic analysis of *GmSGF14* in soybean. (**A**) Collinearity analysis and gene duplication of soybean *GmSGF14* genes. Gray lines represent all homologous gene blocks in soybean genome, and red lines represent duplicated *GmSGF14* gene pairs. (**B**) Phylogenetic tree of 14-3-3 genes in *Arabidopsis*, rice and soybean. Different subgroups are represented by different colors.

**Figure 2 ijms-24-09436-f002:**
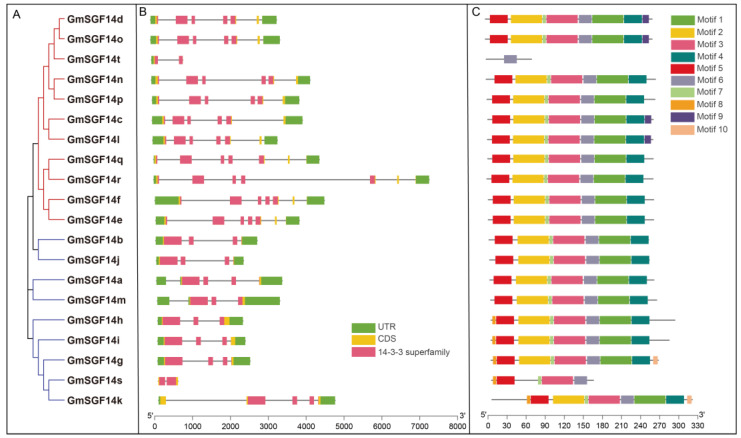
Phylogenetic relationship, gene structure and conserved motifs of *GmSGF14* genes. (**A**) Phylogenetic tree of soybean GmSGF14 proteins, with different subfamilies represented by different colors. (**B**) Genetic structure of *GmSGF14* genes, including introns, UTRs, CDS and domains specific to the 14-3-3 family. (**C**) Conserved motifs of GmSGF14 proteins.

**Figure 3 ijms-24-09436-f003:**
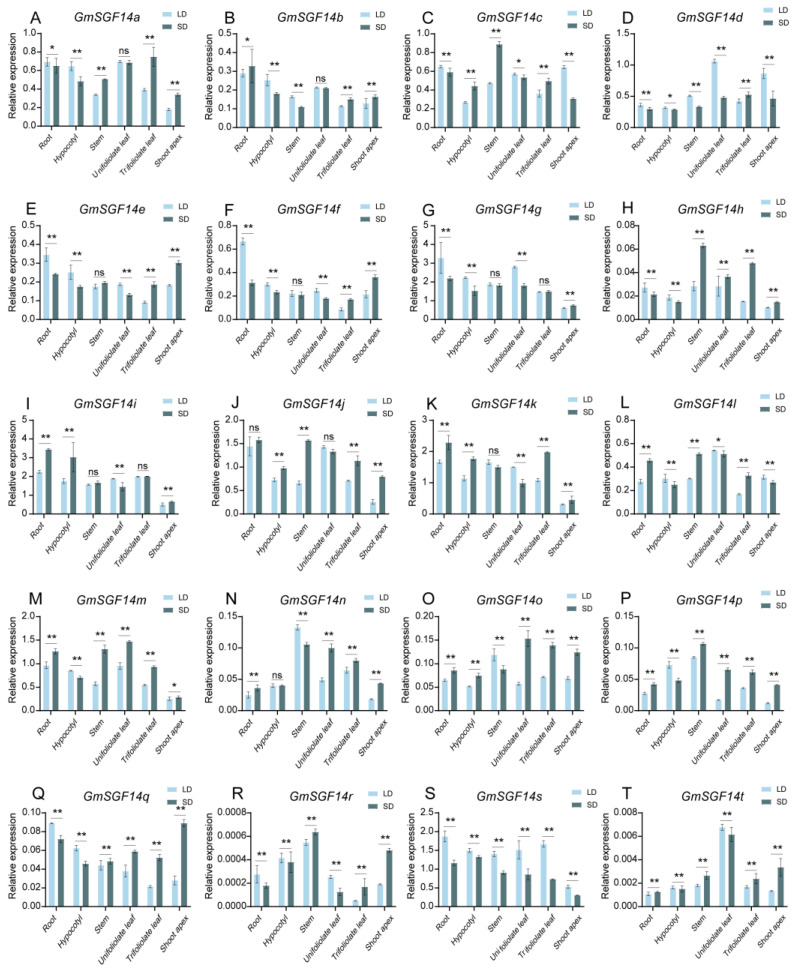
Expression profile analysis of *GmSGF14* in different tissues of soybean variety Zhonghuang 13 under different photoperiod conditions. (**A**–**T**) The expression levels in different tissues of GmSGF14a-GmSGF14t. LD: long day (16: 8 h, light: dark), SD: short day (12: 12 h, light: dark). Error bar represents SE values of three biological replicates. Statistical significance was determined using Student’s *t*-tests: ns, * and ** indicate non-significant results, *p* < 0.05 and *p* < 0.01, respectively.

**Figure 4 ijms-24-09436-f004:**
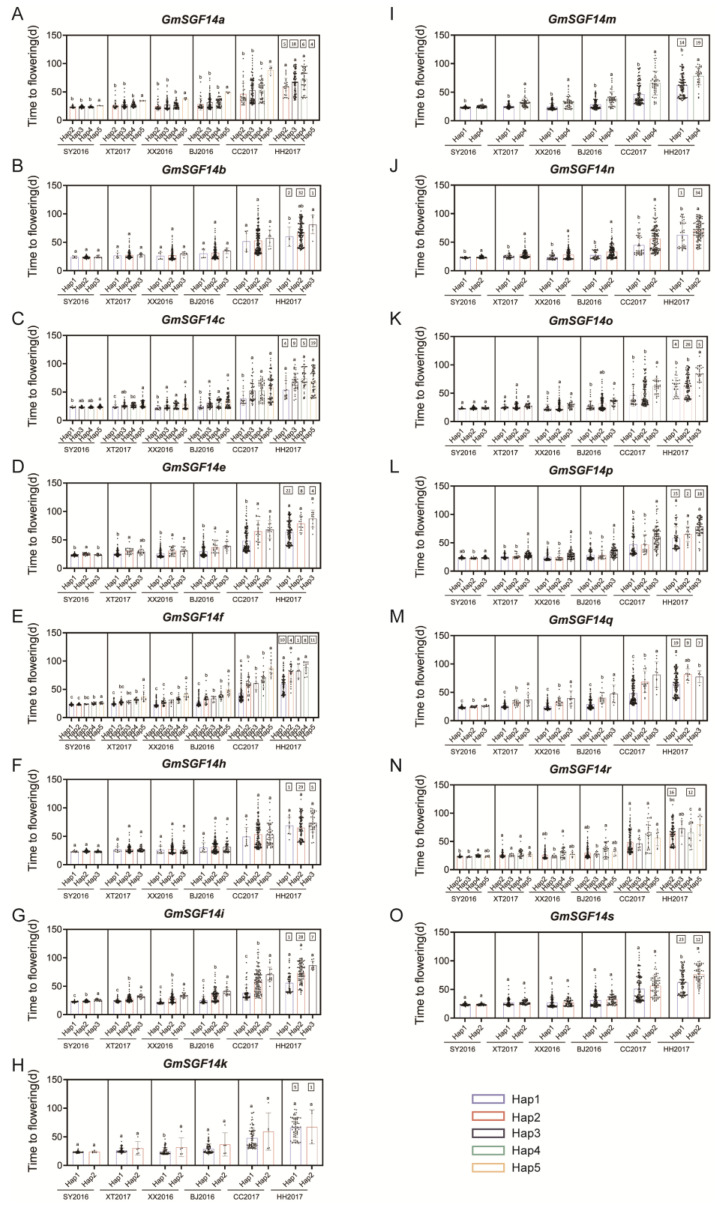
Correlation analysis of *GmSGF14* haplotypes with flowering time in 207 soybean varieties. (**A**–**O**) Flowering time of soybean varieties with major haplotypes of *GmSGF14a*-*GmSGF14s*. Number within each box above bar chart indicates number of unflowering varieties. Data are means ± standard deviations. a-, b- and c-rank determined via Duncan’s test at *p* < 0.05. SY2016: Sanya 2016; XT2017: Xiangtan 2017; XX2016: Xinxiang 2016; BJ2016: Beijing 2016; CC2017: Changchun 2017; HH2017: Heihe 2017.

**Figure 5 ijms-24-09436-f005:**
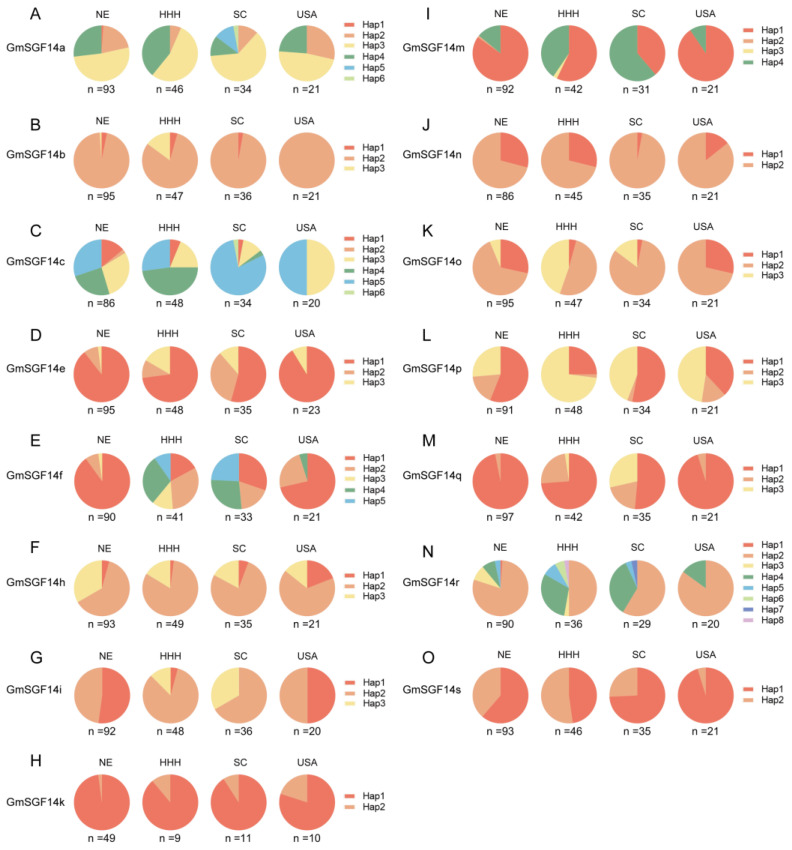
Geographical distribution of soybean varieties with major haplotypes of *GmSGF14*. (**A**–**O**) Geographic distribution of *GmSGF14a*–*GmSGF14s* haplotypes. NE: Northeastern China; HHH: Huang–Huai–Hai; SC: Southern China.

## Data Availability

The sequence data used in this study are available in the NCBI database under Short Read Archive (SRA) Accession Numbers SRP062560 and PRJNA589345.

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
