# Peer review of "Haplotype Analysis of GmSGF14 Gene Family Reveals Its Roles in Photoperiodic Flowering and Regional Adaptation of Soybean"

_ijms, 2023, doi:10.3390/ijms24119436_

Round 1

Reviewer 1 Report

The purpose of Fig. 3 is to show the differences between long days and short days.  Rather than show the color values for each, show only the delta value (the difference between LD and SD).  The original Fig. 3 could be move to a supplementary figure.

Break results section 2.5 into two subsections, one for photoperiod responsive genes and one for non-photoperiod responsive genes.

Some conclusions regarding phenotype are based only on a single environment and single environment conclusions need to be taken lightly. A suggestion is given directly in the text, as add " ... exhibited significantly later flowering in Heihe although this was tested only in a single environment."

Make use of protein-protein interaction prediction tools to see if GmSGF14 genes interact with some of the soybean homologs as you reported for other crops.

Other suggestions are make on the attached version of the manuscript.

The quality of English is generally good.  Some suggestions were made on the attached version of the manuscript. 

Reviewer 2 Report

The article has demonstrated interesting finidngs in soybean in relation to the flowering time and photoperiod sensitivity. These traits are the major ones in determining the soybean adaptation to a wider range of geographic environments. 

The objective of this study was clearly indicated and the research was conducted acordingly to investigate the molecular and evolutionary characteristics of the GmSGF14 gene family and and its roles in controlling the flowering response under diverse photoperiod conditions.

The english languare used in the article are to the best scientific standard and are easy to understand by readers.  In some cases, I suggest to use reported form of the language when results and discussions are presented.
